# Nano-SAR Modeling for Predicting the Cytotoxicity of Metal Oxide Nanoparticles to PaCa2

**DOI:** 10.3390/molecules26082188

**Published:** 2021-04-10

**Authors:** Haihua Shi, Yong Pan, Fan Yang, Jiakai Cao, Xinlong Tan, Beilei Yuan, Juncheng Jiang

**Affiliations:** 1Jiangsu Key Laboratory of Hazardous Chemicals Safety and Control, College of Safety Science and Engineering, Nanjing Tech University, Nanjing 210009, China; shihaihuasafety@njtech.edu.cn (H.S.); 15895934959@163.com (F.Y.); jiakaicao@njtech.edu.cn (J.C.); maochennanchn@163.com (X.T.); yuanbeilei@njtech.edu.cn (B.Y.); ypnjut@126.com (J.J.); 2School of Environment & Safety Engineering, Changzhou University, Changzhou 213164, China

**Keywords:** cellular uptake, metal oxide nanoparticles, cytotoxicity, nano-SAR, norm index descriptors

## Abstract

Nowadays, the impact of engineered nanoparticles (NPs) on human health and environment has aroused widespread attention. It is essential to assess and predict the biological activity, toxicity, and physicochemical properties of NPs. Computation-based methods have been developed to be efficient alternatives for understanding the negative effects of nanoparticles on the environment and human health. Here, a classification-based structure-activity relationship model for nanoparticles (nano-SAR) was developed to predict the cellular uptake of 109 functionalized magneto-fluorescent nanoparticles to pancreatic cancer cells (PaCa2). The norm index descriptors were employed for describing the structure characteristics of the involved nanoparticles. The Random forest algorithm (RF), combining with the Recursive Feature Elimination (RFE) was employed to develop the nano-SAR model. The resulted model showed satisfactory statistical performance, with the accuracy (ACC) of the test set and the training set of 0.950 and 0.966, respectively, demonstrating that the model had satisfactory classification effect. The model was rigorously verified and further extensively compared with models in the literature. The proposed model could be reasonably expected to predict the cellular uptakes of nanoparticles and provide some guidance for the design and manufacture of safer nanomaterials.

## 1. Introduction

In recent years, nanotechnology has been considered as one of the key enabling technologies for global economic growth. With the continuous development of nanotechnology, new kinds of nanomaterials are springing up all over the world [1,2,3]. Nanomaterials are widely used in traditional materials, catalysis [4], medical devices [5,6], electronic equipment [7], coatings, and other industries [8,9,10] owing to their unique properties, such as excellent optical, electrical, and magnetic properties. More and more attention has been paid to the inherent disadvantages of nanomaterials and the resulting hazards that may be exposed in the workplace among consumers and in the environment. Although recent studies have found that some nanomaterials may have biological hazards, understanding of the adverse effects of these products is still in its infancy.

In vitro and in vivo studies are commonly used to assess biological or toxic effects [11]. Nevertheless, experimental methods are laborious, time-consuming, and sometimes involve some ethical issues. Thus, there is a strong desire to build a fast and high-throughput nano-toxicity evaluation system or prediction model as a supplement to traditional experimental methods. Among different kinds of methods, quantitative structure–activity relationship (QSAR) is seen as the most promising approach, which was proposed in the early stages by Corwin Hansch in 1962 and then exploited for developing novel chemicals, primarily for drugs [12]. QSAR is mainly based on the following hypothesis: the molecular structure of a compound contains information that determines its physical, chemical, and biological properties. These physical and chemical properties further affect the biological activity of the compounds. That is to say, an association is found between the molecular structure and biology-related activity of the compounds. A great deal of investigations indicate that it is very urgent and essential to extend the traditional QSAR paradigm to nano-sized materials and evolve “nano-(Q)SAR” models to relate the properties of interest with structure information of novel synthetic nanoparticles, which can provide a theoretical basis for the design of functionalized nanoparticles with expected characteristics [13].

Pancreatic cancer is the fourth leading cause of cancer death with a survival rate of less than 5% at five years. At present, many studies [14,15,16,17] have reported the inhibitory effect of some chemical reagents to pancreatic cancer cells, such as gemcitabine, paclitaxel, and berberine. However, the prognosis is still poor. So far, no chemotherapy has demonstrated efficacy in terms of survival for this cancer. Nanomaterials are increasingly used in daily life, but the safety issues they would cause cannot be ignored, especially their biological toxicity. Due to the intermittent or frequent exposure to the human body, metal oxide nanoparticles (MNPs) may invade the human body through various accessible paths, such as inhalation, skin absorption, and ingestion [18]. Once invaded the human body, they may cause systemic, cellular, or genome toxicity, and of course, it may be exposed to pancreatic cells. Therefore, researches on PaCa2 cell are still necessary.

Currently, different nano-(Q)SAR researches have been conducted for predicting the cellular uptake of 109 functionalized magnetic fluorescent MNPs in PaCa2 cell line. All MNPs possess same superparamagnetic core decorated with different synthetic small molecules [19,20,21]. Chau et al. [22] developed a nano-SAR model for predicting the cellular uptake of 105 nanoparticles to pancreatic cancer cell lines with a single metal core. Four modeling methods were employed to develop candidate models, namely, support vector machine, k nearest neighbor, Logistic Regression and Naïve Bayes. The eventual consensus models had a sensitivity of 86.7 to 98.2% and specificity of 67.3 to 76.6%. Kar et al. [23] developed a more accurate cellular uptake model with six conceptually simple and computable descriptors through partial least squares (PLS) regression approach. Winkler et al. [24] calculated two-dimensional Dragon descriptors, then used linear and nonlinear methods to generate four nano-QSAR models for predicting the uptakes of PaCa2 and human umbilical vein endothelial cell lines (HUVEC). Ojha et al. [25] predicted the uptakes of PaCa2, HUVEC, and human macrophage (U937) cell lines by calculating two-dimensional Dragon descriptors and SiRMS descriptors. Toropov et al. [26] established a reliable nano-QSAR model by using the best descriptor based on SMILES, and then the best parameters were selected using Monte Carlo partial least squares (MC-PLS), 109 datasets were divided randomly into five groups and established QSAR modeling separately. Ronghua Qi et al. [18] developed two nano-QSAR models to predict the cellular uptakes of 109 nanoparticles to PaCa2 and HUVEC cell lines.

In this work, the norm index descriptors were firstly used to describe the structural properties of the MNPs involved. Then, based on the nano-SAR modeling principles of the Organization for Economic Cooperation and Development (OECD) [21], a nano-SAR model was developed to predict the cellular uptake endpoints of 109 MNPs with different surface modification in the PaCa2 cell line. Finally, internal and external verification were made to strictly verify the developed model and define its applicability domain. The model contributes to understand nano-SAR and provide theoretical basis for the design and synthesis of green nanomaterials with high efficiency and harmlessness.

## 2. Results and Discussion

### 2.1. Nano-SAR Model Performance

Based on the data of cellular uptakes of 109 magnetic fluorescent MNPs with surface modification in the PaCa2 line, a nano-SAR model with toxicity endpoint as the dependent variable is established. The performance of the model on the training and test set is assessed with the indicators defined in Equations (1)–(4) [27]. True positive (*TP*) represents that a toxic MNP is correctly classified as positive, true negative (*TN*) represents that a non-toxic MNP is correctly classified as negative, while false positive (*FP*) represents a non-toxic MNP is incorrectly classified as toxic and false Negative (*FN*) represents a toxic MNP is incorrectly classified as non-toxic.
(1)$$SE=\frac{TP}{TP+FN}$$
(2)$$SP=\frac{TN}{TN+FP}$$
(3)$$ACC=\frac{TN+TP}{TN+TP+FP+FN}$$
(4)$$MCC=\frac{TP\times TN-FP\times FN}{\sqrt{\left(TP+FN\right)\left(TP+FP\right)\left(TN+FN\right)\left(TN+FP\right)}}$$

Given the above calculation, the detailed statistical parameters are given in Table 1.

The results of the real label and the predicted label are shown in the confusion matrix in Figure 1.

### 2.2. Model Stability Validation and Results Assessment

The cross-validating process is a statistical method to evaluate the stability of the models. It is more stable and comprehensive than the method of dividing the training set and test set in a single way. In this work, the result of five-fold cross-validating process is 0.909, which demonstrates the good stability and reliability of the model, and can be reasonably used for predicting the cytotoxicity of MNPs.

The ACC of the test set is 0.950, indicating that the classifier has good classification effect and predictive ability, in addition, the subtle difference between the ACC of training and test set (0.966 and 0.950) shows that the model is effective and not subject to overfitting. Furthermore, the results show that the sensitivity and specificity values are greater than 0.9 in the entire data. Fjodorova et al. [28] recommended that the supervised model should be high sensitivity and specificity. It should be noticed that sensitivity is a very significant parameter in a nano-SAR model. Actually, the low sensitivity value indicates the model has a low ability to distinguish the toxicity of various compounds. The specificity is another important indicator. High specificity value means the model has a high ability to distinguish the false positive compounds [29].

The above results show that the model provides high classification accuracy after internal and external verification, and can be reliably employed for predicting the cytotoxicity of MNPs. Moreover, this work indicates that it is possible to predict the cytotoxicity of MNPs through the nano-SAR method using norm index descriptors. Once a reliable model is established, the cytotoxicity of MNPs can be quickly predicted by input of structural parameters of MNPs.

### 2.3. Applicability Domain of the Proposed Model

It should be noted that any developed nano-SAR model should have a clear application domain (AD). As for any nano-SAR model, that only the predictions for materials are within its AD can make it considered to be reliable. In this study, for each category, all test set samples are within the application domain, and the model is reliable.

### 2.4. Comparisons with Other Models in the Literature

The proposed model for predicting cellular uptakes of MNPs in the PaCa2 cell line is based on identical data set reported in the literature. Comparisons of the present model with other reported models for the cellular uptake of MNPs was carried out (shown in Table 2). The external predictability metrics could indicate the prediction performance of proposed models, it was not hard to find that the performance of the present model outperformed those of the previous models proposed by Singh et al. [30]. In particular, it should be noticed that models of Singh et al. were established using eleven descriptors, while only five descriptors were employed in our work. Based on a statistical perspective, the more input descriptors employed in the proposed model, the better statistical parameters will be obtained. Nevertheless, the basic strategy of nano-SAR analysis is to find optimum relationship models between the molecular structures and desired properties with selected descriptors as less as possible. The nano-SAR models with fewer employed descriptors can be considered to be more robust and simpler to use.

## 3. Materials and Methods

### 3.1. Data Set

The dataset of the cellular uptake of 109 nanoparticles was taken from the published article [19] and presented in Appendix A. All nanoparticles had the same metal core with different surface-modifying organic molecules. Nanoparticles were made magnetofluorescent with the addition of fluorescein isothiocyanate (FITC) molecules on their surfaces to enable measurement of cellular uptake. Compared to other cell lines, it was found that the cellular uptake in PaCa2 had more obvious diversity and was highly dependent on surface modifications. Thus in our work, the uptakes data of MNPs in PaCa2 were employed for the model development. Cellular uptake had the expression to be the logarithm of MNPs concentration (pM) in each cell, ranging from 2.23 to 4.44.

For binary classification, the standard of Chau and Yap was referred [22]. Due to this standard, the MNPs achieving cellular uptakes of over 5000 NPs per cell were regarded as better cellular uptakes (positive class), while MNPs with cellular uptakes of less than 5000 particles per cell were regarded as poor cell uptakes (negative class). Therefore, 59 MNPs were in positive class and the end-point values were set at 1, and the rest 50 MNPs were in negative class and the end-point values were set at 0.

### 3.2. Dataset Splitting

Dividing the dataset is an indispensable step for the development of nano-SAR study. Before nano-SAR modeling, all the whole 109 nanoparticles in the data set were randomly divided into a training set with 89 data and a test set with 20 data. The training set is applied to develop the nano-SAR model, whereas the test set is employed for evaluating the performance.

### 3.3. Molecular Descriptors Calculation

Here, we adopted one novel type of norm index descriptors reported by Yali Wang et al. [31] to predict the cellular uptakes of MNPs. The detailed calculating procedures are as follows: Firstly, the 3D structure of each MNP was achieved with Chemdraw (version 14), with the optimization by complying with the MM2 module (the program of class 1 Allinger molecular mechanics). Secondly, for further optimization, the GAUSSIAN (version GAUSSIANVIEW 6.0.16) was employed to carry out Density Functional Theory (DFT) M06-2X functional calculation on the basis of 6-311+G (d, p). Then, a range of distance matrices consisting of step matrix *DM1* and Euclidean distance matrix *DM2* were retrieved from the optimized structures. The specific calculating procedure is as follows:(5)$$DM1=\left[a_{ij}\right]\;\;\left(a=n\;the\;path\;between\;atom\;ij\;is\;n\right)$$
(6)$$DM2=\left[b_{ij}\right]\;\;b_{ij}=\{\begin{array}{c} r_{ij}\\ 0 \end{array}\;\;\;\begin{array}{c} if\;i\neq j\\ if\;i=j \end{array}$$
$r_{ij}$ denotes the Euclidean spatial distance of atom *i* and *j.* Moreover, for introducing the contribution of single atom and enhancing the performance of the approach, here, a property matrix *PM* integrated with several atomic properties was proposed and defined as:(7)$$PM=\left[SN\;EN\;E_{i}\;\tanh\left(ac\right)\right]$$
where, *SN, EN,*
$E_{i}$ and *ac* are electron shell number, electro-negativity, ionization energy, and atom charge, separately.

Next, integrating the matrices *DM* and the proposed property matrix *PM*, three extended matrices were made, and the combinational details are as follows in Equations (4)–(6):(8)$$EM_{1,m,n}=\left[DM_{m}\;\;\;PM\left(:,n\right)\right]$$
(9)$$EM_{2,m,n}=\left[PM\left(:,n\right)\times PM{\left(:,n\right)}^{T}+DM_{m}\right]$$
(10)$${\mathrm{EM}}_{3,m,n}=\left[\left(PM\left(:,n\right)\times PM{\left(:,n\right)}^{T}\right)\times DM_{m}\right]$$

With these matrices, we employed the norm indexes consisting of *norm (EM, 1)*, *norm (EM, 2)* and *norm (EM, 3)*. Herein, *norm (EM, 1)*, *norm (EM, 2)* and *norm (EM, 3)* refer to the largest column sum, the largest singular value, and the Frobenius norm of the matrix *EM*, separately. Therefore, three norm indexes have the definition as Equations (7)–(9):(11)$$norm\left(EM,1\right)=\underset{j}{\max}\left[\overset{p}{\underset{i=1}{\sum }}\left|EM_{ij}\right|\right]\;\;j=1,\ldots ,q$$
(12)$$norm\left(EM,2\right)=\sqrt{\max\left(\lambda_{1}\left(EM^{H}\times EM\right)\right)}$$
(13)$$norm\left(EM,3\right)=\sqrt{\left(\overset{q}{\underset{j=1}{\sum }}\overset{p}{\underset{i=1}{\sum }}EM_{ij}^{2}\right)}$$
where *p* and *q* are the number of rows and columns of matrix *EM*, respectively. The $\lambda_{i}$ refers the eigenvalue of the matrix. The $EM^{H}$ refers the Hermite matrix of the matrix *EM*.

### 3.4. Descriptor Selection and Modeling

High-dimensional data will not only increase the complexity of calculation, but also lower the efficiency of the predictive models for classification [32]. In order to establish an effective and reliable model, it is, therefore, essential to select the most relevant features. In this study, we decreased the dimension of feature space using the Random forest algorithm (RF), combining it with the Recursive Feature Elimination (RFE) [33], which could eliminate data redundancy and generate more compact feature subsets. Figure 2 illustrates the process of the RF-RFE approach. Firstly, we used the RF algorithm to train our model by complying with the training data, and the importance of each feature was obtained based on the relevant classification contribution. Then, the features were sorted based on their importance from high to low. A ranking of features was obtained here. Finally, we eliminated the least important feature, and then retrained the RF model with the updated features, and obtained the classification performance with the current feature set. This is an iterative process until the feature set is empty. As a result, a list of performance measurement values corresponding to each subset was generated. All these steps above were carried out by PyCharm software (PyCharm Community Edition 2019.3.4).

### 3.5. Model Validation

Model validating process can be absolutely necessary for ensuring reliability of the developed nano-SAR model. According to the OECD regulations [21], only validated models can be considered to be reliable. Here, we adopted all kinds of validating methods to validate the performance of the developed nano-SAR model for its fitness, robustness, and predictability.

Firstly, for binary classification, the most commonly used statistical parameters such as Sensitivity (SE), Specificity (SP), Accuracy (ACC), and Matthews correlation coefficient (MCC), were used to evaluate the fitness of the nano-SAR model [34].

Secondly, the robustness of the model was represented by the k-fold cross-validating process (k-CV), k usually takes five or ten, which is the most common method in the internal validating process [35,36]. The advantage of this method is that it can perform reliable and fair testing on the dataset [37]. In this way, not only the robustness but also the internal predictability of the model can be verified.

In addition, the nano-SAR model is often validated in two steps, that is, the internal validating process and the external validating process. The external validating process is fairly significant and widely used method to evaluate both the external predictability and the generalizability of the nano-SAR model for novel compounds. Here, the external validating process was executed by splitting the available data set into a training set and an external test set. The training set is used for selecting descriptors and developing models, while the test set is used to achieve external validation.

### 3.6. Applicability Domain (AD)

According to the OECD standard 3rd, it is necessary to determine the application domain of the model when an acceptable (Q)SAR model is proposed. AD describes the physicochemical space upon which the developed model is trained, and thus can be applied to make predictions. Merely the structures of the new compounds are “similar” to those in the training set can obtain an effective prediction result [38]. That is, for each category (toxic and non-toxic) in this study, if the leverage value of the test set sample is within the range of the training set, the prediction is considered to be valid. Otherwise, it is considered to be beyond the application domain of the model, the prediction result is invalid. The leverage value *h_i_* is defined as: $h_{i}=x_{i}^{T}{\left(X^{T}X\right)}^{-1}x_{i}$, where *x_i_* denotes a row vector of descriptors for a particular *i*th MNP and *X* denotes the m × n matrix of descriptors in all samples.

## 4. Conclusions

In this work, a new nano-SAR model based on norm index descriptors was developed to predict the cytotoxicity of 109 functional magnetic fluorescence MNPs to the PaCa2 cell line. The results indicate that the developed model could provide satisfactory predictions. Based on several internal and external validating strategies, the robustness and predictivity of the model were rigorously validated. The main findings of this study include:The employed norm index descriptors combining the atomic distance matrices with the property matrix could accurately and effectively characterize the structural features of MNPs and lead to a nano-SAR model with satisfactory model performance.The Random forest algorithm (RF) combined with the Recursive Feature Elimination (RFE) method could be successfully employed to explore and describe the internal relationships between the nanostructure and cytotoxicity of MNPs.Since a considerable number of MNPs were involved in the development of the model, and a rigorous model validating process and extensive model comparisons were performed, the proposed model in this study could be reasonably considered as reliable in predicting the cytotoxicity of novel MNPs or other MNPs for which experimental data are unknown.

## Figures and Tables

**Figure 1 molecules-26-02188-f001:**
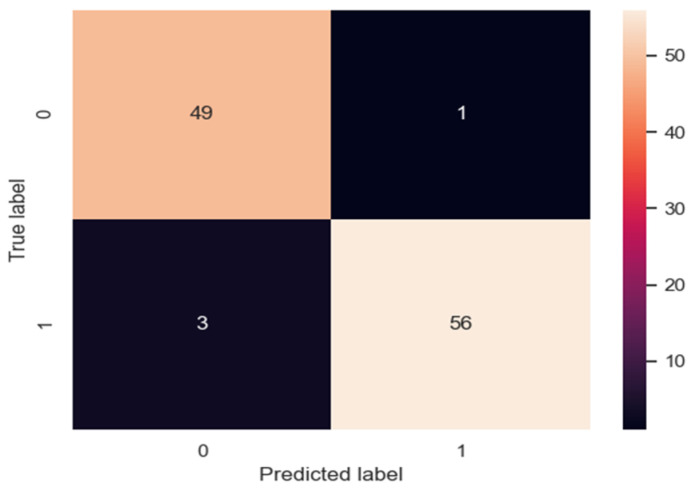
Confusion matrix.

**Figure 2 molecules-26-02188-f002:**
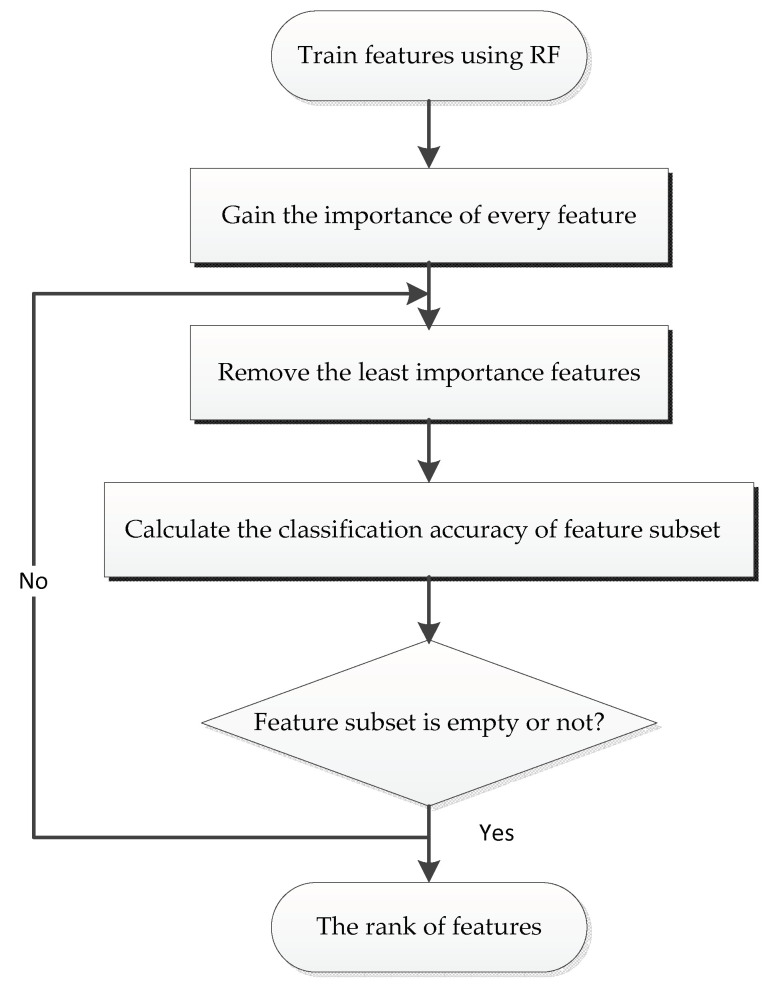
The main procedure of the recursive feature elimination (RFE) method.

**Table 1 molecules-26-02188-t001:** Performance matrices of the full model.

Sub-Set	*n*	SE	SP	ACC	MCC
Training set	89	0.958	0.976	0.966	0.933
Test set	20	0.909	1	0.950	0.905
Complete	109	0.949	0.980	0.972	0.927

**Table 2 molecules-26-02188-t002:** Comparison of statistical parameters between present model and past models.

Works	Method	Sub-Set	SE	SP	ACC	MCC
Singh et al.	DTB	Training set	1	0.974	0.988	0.980
		Test set	0.882	1	0.926	0.860
	DTF	Training set	1	1	1	1
		Test set	0.875	0.909	0.889	0.780
This work	RF	Training set	0.958	0.976	0.966	0.933
		Test set	0.909	1	0.950	0.905

## Data Availability

The data presented in this study are available in Appendix A.

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
