# Peer review of "Nano-SAR Modeling for Predicting the Cytotoxicity of Metal Oxide Nanoparticles to PaCa2"

_molecules, 2021, doi:10.3390/molecules26082188_

Round 1
Reviewer 1 Report
The authors refer that "The dataset of the cellular uptake of 109 nanoparticles on one predominant metal core platform but bearing different organic coatings (small organic molecules) was taken from the published article [17]and presented in Table S1." This article is not OpenAccess and the authors present NO permission to reuse the data from Cell-specific targeting of nanoparticles by multivalent attachment of small molecules owned by Springer Nature, Nature Publishing Group. And the authors do not refer correctly to the creator and the owner of the initial dataset.
Otherwise, it may be considered as data-mining or similar. Till this issue is not clearly clarified, the paper cannot be reviewed. However, the language is complicated and not easily readable.
The introduction is not related to the methods the study uses as the authors did not study any toxicity or used any cell-line. There is no explanation of why they pick up the data generated using pancreatic cell-line as pancreatic cells are not epithelial cells and the exposure of pancreatic cells is much unusual and needs to be compared with the datasets of other origins to be metaanalysis.
Reviewer 2 Report
Comments and Suggestions for Authors
The manuscript entitled "Nano-SAR modeling for predicting the cytotoxicity of metal
3 oxide nanoparticles to PaCa2" develops a new nano-SAR model based on norm index descriptors predict the cytotoxicity of functional magnetic fluorescence MNPs on the PaCa2 cell line. The results indicate that the developed model could provide satisfactory predictions.
The manuscript is well-written and the finding is novel. Therefore it is appropriate that manuscript can be published in Molecules after revision.
I have the following comments on the manuscript
- The manuscript needs minor English language revision/editing.
- Ensure the rational of work
Round 2
Reviewer 1 Report
Thank the authors for improving the paper.
Please, could You kindly correct the language. For example
row 77: predict the cellular uptakes of 109 datasets to PaCa2 and HUVEC lines. Do I correctly understand that the datasets were uptaken by cell-lines?
row 83: in the PaCa2 line or by the PaCa2 cell-line
etc. similar
row 92: Nanoparticles were made into magnetic fluorescence or nanoparticles were fluorescence-labelled by
row 254 it must be pay attention or attention should be paid
row 278 Since considerable MNPs were involved or considerable amount?
Regarding “There is no explanation of why the authors pick up the data generated using pancreatic cell-line as pancreatic cells are not epithelial cells and the exposure of pancreatic cells is much unusual” I do express that the initial paper from where the data was taken is rather old and the use of pancreatic cells is not justifiable as those cells do not express the properties of defence barrier of the organism although “uptake by PaCa2 had more obvious diversity and was highly dependent on surface modifications”. In real conditions that model may not be usable as epithelial cells act differently. If the model created by authors is good to use it should be effective as well with other data sets collected using contemporary quality validation and reasonably chosen cell-line.
